# Carbon Nanotubes Interconnected NiCo Layered Double Hydroxide Rhombic Dodecahedral Nanocages for Efficient Oxygen Evolution Reaction

**DOI:** 10.3390/nano12061015

**Published:** 2022-03-20

**Authors:** Meng Li, Yujie Huang, Jiaqi Lin, Meize Li, Mengqi Jiang, Linfei Ding, Dongmei Sun, Kai Huang, Yawen Tang

**Affiliations:** 1School of Chemistry and Chemical Engineering, Southeast University, Nanjing 211189, China; limeng374@163.com; 2Jiangsu Key Laboratory of New Power Batteries, Jiangsu Collaborative Innovation Centre of Biomedical Functional Materials, School of Chemistry and Materials Science, Nanjing Normal University, Nanjing 210023, China; huangyj0206@163.com (Y.H.); linjq0215@163.com (J.L.); meitselee@163.com (M.L.); jiangmengqi666@163.com (M.J.); sundongmei@njnu.edu.cn (D.S.); 3Advanced Analysis and Testing Center, Nanjing Forestry University, Nanjing 210037, China; jsdinglinfei@163.com

**Keywords:** NiCo layered double hydroxides, carbon nanotubes, self-sacrificing template, hollow structure, oxygen evolution reaction

## Abstract

Proper control of a 3*d* transition metal-based catalyst with advanced structures toward oxygen evolution reaction (OER) with a more feasible synthesis strategy is of great significance for sustainable energy-related devices. Herein, carbon nanotube interconnected NiCo layered double hydroxide rhombic dodecahedral nanocages (NiCo-LDH RDC@CNTs) were developed here with the assistance of a feasible zeolitic imidazolate framework (ZIF) self-sacrificing template strategy as a highly efficient OER electrocatalyst. Profited by the well-fined rhombic dodecahedral nanocage architecture, CNTs’ interconnected characteristic and structural feature of the vertically aligned nanosheets, the as-synthesized NiCo-LDH RDC@CNTs integrated large exposed active surface areas, enhanced electron transfer capacity and multidimensional mass diffusion channels, and thereby collaboratively afforded the remarkable electrocatalytic performance of the OER. Specifically, the designed NiCo-LDH RDC@CNTs exhibited a distinguished OER activity, which only required a low overpotential of 255 mV to reach a current density of 10 mA cm^−2^ for the OER. For the stability, no obvious current attenuation was detected, even after continuous operation for more than 27 h. We certainly believe that the current extraordinary OER activity combined with the robust stability of NiCo-LDH RDC@CNTs enables it to be a great candidate electrocatalyst for economical and sustainable energy-related devices.

## 1. Introduction

The oxygen evolution reaction (OER, 4OH^−^ → 2H_2_O + O_2_ + 4*e*^−^), as one of the most fundamental energy-intensive anodic reactions, plays an important role in many energy-related sustainable systems such as water-splitting devices [1,2,3], fuel cells [4,5,6,7,8], rechargeable metal–air batteries [9,10,11], and sustainable CO_2_ reduction systems [12,13,14]. Unfortunately, the decisive OER process suffers from its intrinsically sluggish reaction kinetics due to their complicated multi-step proton-coupled electron transfer paths [15]. As such, the OER process comprises the breakage of the O–H bond and the concomitant formation of the O=O double bond, which brings about the high thermodynamically uphill reaction energy barriers of OER (237 kJ mol^−1^) [16]. Consequently, rational construction of high-efficiency electrocatalysts to expedite the flat-footed OER process is of great significance. Up until now, noble metal Ru/Ir-based oxides have been deemed as the benchmark electrocatalysts for the OER. However, their practical applications are jeopardized by the rarity, prohibitive cost, and inferior electro-stability. Therefore, it is still far-reaching and challenging to explore both economical and efficient alternative electrocatalysts to accelerate the OER process.

During the past few decades, a myriad of 3*d*-transition metal-based catalysts have been dedicated to promoting the OER process such as oxides [17,18], phosphides [19], alloys [20], hydroxides [21,22], sulfides [23], nitrides [24,25], etc. Since Dai et al. first reported that NiFe layered double hydroxides (LDHs) exhibited high catalytic activity toward water oxidation in 2013 [26], a large number of studies have been conducted to improve its catalytic performance to meet the requirements of large-scale applications. Among many alkaline OER catalysts, 3*d*-trantion metal based LDHs usually possess lower OER overpotential and smaller Tafel slope than perovskites, alloys, metal oxides, etc. Generally, LDH is a unique classic term of two-dimensional (2D) hydrotalcite and hydrotalcite-like intercalation materials with the general structural formula of [M^II^_1−*x*_M^III^*_x_*(OH)_2_]*^x^*^+^[A*_x_*_/*n*_]*^n^*^−^·*y*H_2_O [27]. In this structure, the bivalent/trivalent metal ions can be easily replaced by other metal ions while the interlayer anions and water molecules stabilize the structure by electrostatic interactions and hydrogen bonds [28]. Thus, this flexible formula feature enables LDHs with a tunable composition and electronic configuration, which is extremely important for the electrocatalytic process. However, the bulk LDHs ordinarily possess instinctual poor electron conductivity and inherent lamellar structure, which can hardly afford sufficient active sites, greatly hinder the electron transfer process (electrode–catalyst–reactant), and tend to severe agglomerations during the OER process [29]. To overcome the above-mentioned problems, the engineering architecture (hollow nanocage structure, hierarchical arrayed structure, etc.) and introduction of a conductive connector seem to be two reliable strategies to enhance the OER performance of LDHs. Many reports have shown that the catalytic properties of nanomaterials are closely related to the exposure of surface catalytic activity centers [24,25]. Architecturally, the morphological engineering of LDHs (shape and size, porosity, etc.) can efficiently provide more accessible catalytically active sites, tapping luxuriant mass diffusion channels and robust structural and electrocatalytic stability, which will undoubtedly expedite the OER kinetics [30]. Furthermore, the construction of a specific structure such as a hollow nanocage structure can also provide independent reaction chamber/cells, and thus produce more low coordination atoms, promote the mass transfer process, and accelerate reaction rates toward the OER [31]. Nevertheless, as for the introduction of a conductive connector, the intimate connection between the conductive substrate and active components can induce delicate electronic structure modification and charge redistribution, which effectively regulate the adsorption energy of intermediates and improve the electronic conductivity of electrocatalysts [15,32]. However, it is still far-reaching but challenging to realize elaborate control over the above two strategies in one facile and efficient approach.

Herein, we designed and exploited a facile and feasible zeolitic imidazolate framework (ZIF) self-sacrificing template strategy to synthesize carbon nanotube interconnected NiCo layered double hydroxide rhombic dodecahedral nanocages (NiCo-LDHs RDC@CNTs) as a highly efficient OER electrocatalyst. The ZIF-67 templates were first immobilized and interconnected by carbon nanotubes (CNTs) through a mild hydrothermal reaction at room temperature. Afterward, the inner ZIF-67 self-sacrificing templates were gradually dissolved and transformed into ultrathin NiCo-LDHs shells around the rhombic dodecahedral template and CNTs via the hydrolysis of Ni(II) nitride. Benefitting from the well-fined vertically aligned nanosheet assembled rhombic dodecahedral nanocage structure and the CNTs’ interconnected characteristic, the as-prepared NiCo-LDH RDC@CNTs displayed excellent electrocatalytic performance toward OER with a low overpotential of 255 mV at a current density of 10 mA cm^−2^ and excellent long-term stability (continuous operation for more than 27 h without obvious current attenuation).

## 2. Materials and Methods

### 2.1. Reagents and Chemicals

Cobalt nitrate hexahydrate (Co(NO_3_)_2_·6H_2_O), cobalt hydroxide, 2-methylimidazole, and ruthenium oxide (RuO_2_) were purchased from Aladdin Biochemical Technology Co. Ltd. (Shanghai, China). PVP-K30 was purchased from Beijing Solarbio Science & Technology Co. Ltd. (Beijing, China). CNTs were obtained from Nanjing XFNANO Materials Tech Co. Ltd. (Nanjing, China). Methanol, ethanol, and nitric acid (HNO_3_) were bought from Sinopharm Chemical Reagent Co. Ltd. (Shanghai, China).

### 2.2. Methods

For the pretreatment of CNTs, 30 mg CNTs was dissolved in 40 mL 3 M nitric acid with continuous stirring for 6 h. The processed CNTs were collected by centrifugation, washed several times with deionized water and dispersed in 6 mL ethanol.

For the synthesis of ZIF-67@CNTs, 1 mL CNT ethanol solution was dispersed in 19 mL ethanol with continuous ultrasound for 2 h. Subsequently, 1.0 g PVP was added to the above solution and stirred for 15 min. Then, 5 mL 0.5 M Co(NO_3_)_2_ methanol solution and 5 mL 0.5 M 2-methylimidazole (2-MIM) methanol solution were added to the obtained mixture, respectively, and stirred for 12 h. Finally, the precipitate was collected by centrifugation, washed several times with ethanol, and then dispersed in 3 mL ethanol. For comparison, ZIF-67 was synthesized via the similar procedure except without the introduction of CNTs.

For the synthesis of NiCo-LDHs RDC@CNTs, 5 mL ZIF-67@CNT ethanol solution was dissolved in 45 mL ethanol with continuous ultrasonic dissolving for 30 min. Then, 90 mg Ni(NO_3_)_2_·6H_2_O was dissolved in the above solution by magnetic stirring for 5 min. Subsequently, the reaction mixture was stirred and reacted at 40 °C for 4 h. Finally, the precipitate was collected by centrifugation, washed several times with ethanol, and then dried at 40 °C overnight.

### 2.3. Characterization

The X-ray powder diffraction (XRD) was carried out on a D/max-rC X-ray diffractometer with Cu Kα radiation (λ = 1.5406 Å) to investigate the crystallinity of the samples. To examine the morphologies of the samples, transmission electron microscopy (TEM) images and scanning electron microscope (SEM) images were captured on a JEOL JEM-2100F with an accelerating voltage of 200 kV and JEOL JSM7500F (Tokyo, Japan), respectively. The high-angle annular dark-field scanning TEM (HAADF-STEM, Tokyo, Japan) images and energy dispersive X-ray spectroscopy (EDS) elemental mapping/line scan were performed on a JEOL-2100F FETEM at 200 KV. X-ray photoelectron spectroscopy (XPS) was carried out using a Thermo VG Scientific ESCALAB 250 spectrometer (Waltham, MA, USA).

### 2.4. Electrochemical Measurements

All electrochemical tests were carried out in an O_2_-saturated 1.0 M KOH electrolyte by a typical three-electrode system on a computer-controlled CHI760E electrochemical workstation. A catalyst-modified nickel foam (NF) electrode (1 × 2 cm^2^, thickness: 0.3 mm), a carbon rod, and a saturated calomel electrode (SCE) were used as the working electrode, auxiliary electrode, and reference electrode, respectively. For the preparation of catalyst ink, 5 mg of catalysts were dissolved in 450 µL ethanol, 450 µL deionized water, and 100 µL Nafion (10%) mixture solution under continuous ultrasonic treatment for 1 h. For the preparation of NF electrode, 200 µL of the above catalyst ink was used to modify the NF electrode. For the OER, the active area of the NF in the electrolyte is about 1.0 cm^2^. All potentials in this work were converted to the reversible hydrogen electrode (RHE) by the equation E_RHE_ = E_SCE_ + 0.0592 × pH + 0.242. Electrochemical impedance spectroscopy (EIS) measurements were carried out at fixed voltage 0.5 V vs. SCE in the frequency range of 100 kHz to 0.01 Hz.

### 2.5. Zn–Air Battery Test

The Zn–air battery tests were operated by an assembled Zn–air battery. Typically, a 0.3 mm polished Zn plate is employed as anode, while a catalyst modified hydrophilic carbon paper is used as the air-cathode. The catalyst layer was prepared by coating catalyst ink (40 mg mL^−1^) onto the water-facing side of carbon paper with the uploading mass of 5 mg cm^−2^. The active area of the air-cathode was 1 cm^2^. A total of 0.2 M ZnCl_2_ and 6 M KOH mixed solution was used as the electrolyte. The galvanostatic charge–discharge curves were conduct by a Land CT2001A system at the current density of 5 mA cm^−2^, and each discharge/charge cycle was set to be 20 min.

## 3. Results and Discussion

The synthesis procedure of NiCo-LDHs RDC@CNTs is illustrated briefly in Figure 1. First, the ZIF-67 templates were immobilized and interconnected by carbon nanotubes (CNTs) through a feasible and mild one-step hydrothermal reaction under room temperature, where the Co(NO_3_)_2_ and 2-MIM served as precursors while CNTs were used as substrates and the connectivity interconnector. After the hydrolysis process of Ni(NO_3_)_2_ at 40 °C, the inner ZIF-67 templates were gradually dissolved and transformed into NiCo-LDHs. Moreover, the etching process by Ni^2+^ also released free Co^2+^ ions, which co-precipitated with Ni^2+^ and formed NiCo-LDH nanosheets on the surface of the rhombic dodecahedral ZIF-67 template and CNTs. Finally, the inner ZIF-67 templates were completely dissolved and transformed into NiCo-LDH rhombic dodecahedral nanocages without extra template removal steps. The detailed synthesis procedure is described in the Materials and Methods section.

First, the crystalline structure of the as-prepared NiCo-LDH RDC@CNTs was investigated by X-ray powder diffraction (XRD) technology. As shown in the XRD pattern (Figure 2a), the as-prepared NiCo-LDH RDC@CNTs exhibited a typical set of identified peaks, which could be well matched to thee hexagonal phased NiCo-LDH (JCPDF No. 40-0216), indicating the high purity and the LDH crystal structure of the as-prepared catalyst. Moreover, the surface composition and chemical states of NiCo-LDH RDC@CNTs were explored by X-ray photoelectron spectroscopy (XPS). As demonstrated by the survey scan XPS spectrum (Figure 2b), the distinct signals of Ni 2*p*, Co 2*p*, O 1*s*, and C 1*s* could be well detected, indicating the co-existence of the above-mentioned Ni, Co, C, and O elements. The high-resolution Co 2*p* spectrum in Figure 2c could be well disassembled to two sets of spin-orbit doublets and a set of associated shake-up satellite peaks. Specifically, the two main peaks located at binding energies of 781.2 and 796.5 eV well corresponded to the Co^3+^ species, while the two relatively weak peaks centered at 783.6 and 797.7 eV could be attributed to Co^2+^ [9]. Noticeably, based on the peak area quantification at 2*p* 3/2, the atomic ratio of Co^3+^/Co^2+^ was calculated to be about 4.10. Based on previously reported studies, the high content percentage of high-valance-state Co^3+^ species can accelerate the adsorption energy of oxygen intermediates and result in the promotion of OER kinetics [33]. Similarly, the high-resolution Ni 2*p* spectrum (Figure 2d) can also be deconvoluted into Ni^2+^ species (855.4 and 873.1 eV), Ni^3+^ species (856.8 and 874.8 eV), and associated shake-up satellites (861.7 and 879.8 eV) [25]. In the crystalline structure of LDH, the flexible replaceability of bivalent/trivalent metal ions (e.g., Ni^2+/3+^, Co^3+/2+^) can easily result in rich oxygen vacancies and defects in the crystal structure of LDH [34]. As presented by the high-resolution O 1*s* spectrum (Figure 2e), except for the peaks of oxygen in the OH^-^ species (O2, 531.3 eV) and lattice oxygen in M–O bond (O3, 532.1 eV), a certain number of signals of oxygen vacancies (O1) could also be detected at the binding energy of 530.6 eV, validating the high oxygen vacancies content in the as-prepared NiCo-LDH RDC@CNTs [10]. As for the high-resolution C 1*s* spectrum in Figure 2f, three typical peaks located at 284.6, 285.6, and 288.5 eV could be deconvoluted, which could be well assigned to *sp^2^*-hybrided C–C, C–N, and C–O species, respectively [35].

The morphology and crystalline structure of the pre-precursors were investigated in detail with scanning electron microscopy (SEM), transmission electron microscopy (TEM), and XRD. First, the morphology of separate CNTs and ZIF-67 were explored through typical SEM images. As displayed in Figure 3a, the CNTs exhibited an intertwined nanotube structure with an average thickness of about 15 nm. Moreover, the ultra-long characteristics of carbon nanotubes (about tens of microns) could ensure the effective connection of several ZIF-67 templates when serving as a conductive connector. XRD pattern (Appendix A) displays two broad diffraction peaks centered at about 26.5° and 44.0°, which can be well indexed to the (002) and (101) facets of graphitic carbon, respectively. As for the ZIF-67 templates, the SEM image (Figure 3b) showed a typical and well-defined rhombic dodecahedron structure of which the average size of separate ZIF-67 crystal grains was calculated to be about 1–2 µm. The XRD profile (Appendix A) displayed a typical set of diffraction peaks of ZIF-67, indicating the successful synthesis of ZIF-67 crystals. Subsequently, the morphological structure of ZIF-67@CNTs was investigated in detail by TEM technology. In the reaction process, the surface of nitric acid treated CNTs is rich in anionic functional groups (–NO_2_, –OH, etc.), which can effectively adsorb and anchor Co^2+^ ions and then serve as coordination and nucleation sites with 2-MIM to in situ grow ZIF-67 grains on CNTs. As revealed in the large scale TEM image in Figure 3c, the ZIF-67 templates still inherited the well-defined rhombic dodecahedron structure. Nevertheless, in the ZIF-67@CNT sample, the CNTs penetrated throughout the ZIF-67 grains and connected them into strings. Magnified TEM images of an individual CNTs penetrated ZIF-67 crystal (Figure 3d,e) clearly showed that the CNTs were tightly penetrated across the ZIF-67 with a strong interface interaction. The high-angle annular dark-field scanning TEM (HAADF-STEM) image and corresponding element dispersive spectroscopy (EDS) mapping profiles (Figure 3f and Appendix A) also proved the CNTs’ interconnected structure feature and the successful formation of ZIF-67. Furthermore, the XRD pattern of the ZIF-67@CNT sample (Appendix A) displayed a similar result compared to ZIF-67, further indicating the formation of ZIF-67 crystals.

After the hydrolysis etching process by Ni^2+^, the NiCo-LDH RDC@CNTs were finally observed. During this etching process, the protons generated from Ni^2+^ hydrolysis can attack the coordination bond between Co^2+^ and 2-MIM, releasing free Co^2+^, which can react and co-precipitate with Ni^2+^ on the surface of ZIF-67 precursors and CNTs [36]. The precise structure of NiCo-LDH RDC@CNTs were investigated by HADDF-STEM images and TEM images. As shown in the HADDF-STEM images (Figure 4a,b), the as-synthesized NiCo-LDH RDC@CNTs revealed a typical interconnected 3-dimensional (3D) hierarchical nanocage structure. TEM images (Figure 4c) showed that the smooth surface of ZIF-67 precursors were transformed to layers of hierarchical NiCo-LDH nanosheets in randomly oriented arrays, while the inner ZIF-67 templates completely vanished due to the Kirkendall effect [37]. Magnified TEM images (Figure 4d,e) showed that the loosely interconnected LDH nanosheets supported each other and self-assembled into highly porous nanocage architecture. Moreover, the intertwined CNTs were covered with NiCo-LDH nanosheets and penetrated across the typical hollow rhombic dodecahedral nanocages (Figure 4e,f). Benefiting from the unique nanosheet assembled nanocage structural feature, the as-prepared NiCo-LDH RDC@CNTs possessed multi-dimensional mass diffusion channels, more exposed and accessible active sites, and robust structural and mechanical strength, which is crucial to the fast OER kinetics [38]. The N_2_ adsorption–desorption isotherms (Appendix A) of NiCo-LDH RDC@CNTs disclosed the representative IV-type characteristic with a H_3_-type hysteresis loop, proving the mesoporous feature of the as-synthesized NiCo-LDHs RDC@CNTs, which corresponded well with the pore size distribution profile (Appendix A). Due to the unique architecture features, the NiCo-LDH RDC@CNTs displayed a high Brunauer–Emmett–Teller (BET) specific surface area of around 89.9 m^2^ g^−1^. Nevertheless, the intimate interface connections between conductive CNTs and active NiCo-LDH assembled nanocages can link the independent reaction chambers/cells, induce delicate electronic structure modification, serve as electron transfer expressways, and thus accelerate the electron conductivity of NiCo-LDH RDC@CNTs [39]. As for the HRTEM, a set of legible lattice fringes of NiCo-LDH RDC@CNTs were calculated to be about 0.23 nm, which well corresponded to the (015) facets of hexagonal-phased NiCo-LDH (JCPDS No. 40-0216). To explore the element distribution states in the NiCo-LDH RDC@CNTs, element mapping and line scan analysis in STEM mode were carried out on individual CNT penetrated NiCo-LDH RDC. As shown in Figure 2h and Appendix A, it could be clearly observed that the Ni, Co, O, and C elements were uniformly distributed throughout the NiCo-LDH framework, indicating the successful formation of the NiCo-LDH RDC@CNTs, which was well consistent with the EDS results (Appendix A). Furthermore, the EDS line scan profiles (Appendix A) also validated the co-existence of Ni, Co, O, and C elements and the nanocage architecture of NiCo-LDH RDC@CNTs.

To probe the electrocatalytic OER performance of the as-prepared NiCo-LDH RDC@CNTs, a standard three-electrode evaluation system was employed in an O_2_-saturated 1.0 M KOH electrolyte. The basic configuration of the OER electrolyzer is illustrated in the inset of Appendix A. All potentials involved in this section were *iR*-corrected and referenced to the reversible hydrogen electrode (RHE). The OER activity of NiCo-LDH RDC@CNTs were first explored by controlling the reaction temperature. As depicted in Appendix A, the linear sweep voltammetry (LSV) curves indicated that the NiCo-LDH RDC@CNTs with an appropriate synthesis temperature (40 °C) exhibited the best electrocatalytic activity for the OER. For comparison, the OER performance of the ZIF-67@CNT intermediate, ZIF67, Co(OH)_2_, and commercial RuO_2_ counterparts were also evaluated under the similar conditions. As depicted in polarization curves (Figure 5a), the as-prepared NiCo-LDH RDC@CNTs exhibited an extremely high OER activity that delivered a small overpotential of only 255 mV at the current density of 10 mA cm^−2^. Moreover, with the increase in the operation potential, the current density of NiCo-LDH RDC@CNTs exhibited a steep upward trend, representing the superiority of NiCo-LDH RDC@CNTs for the OER. When compared to the other counterpart samples, the NiCo-LDH RDC@CNTs displayed the highest current response among the entire measured potential range, which even exceeded that of the state-of-art RuO_2_ catalyst. To be specific, a set of overpotential comparison histograms was furnished based on the LSV curves. As can be clearly observed in Figure 5b, the NiCo-LDH RDC@CNTs delivered the lowest overpotential of 255 mV to offer the current density of 10 mA cm^−2^ (*E_j_*_10_), which was noticeably lower to that of the ZIF-67@CNT intermediate (*E_j_*_10_ = 336 mV), ZIF-67 (*E_j_*_10_ = 411 mV), Co(OH)_2_ (*E_j_*_10_ = 351 mV), and commercial RuO_2_ (*E_j_*_10_ = 331 mV), indicating that the as-synthesized NiCo-LDH RDC@CNTs could efficiently accelerate the OER process. Apart from the OER activity, good OER kinetic is also extremely important for an ideal OER electrocatalyst. Regarding this point of view, Tafel plots were employed to appraisal the OER kinetics of these catalysts. As displayed in Figure 5c, the Tafel slope of the as-synthesized NiCo-LDH RDC@CNTs was calculated to be about 78.23 mV Dec^−1^, which was smaller than that of the ZIF-67@CNT intermediate (95.42 mV dec^−1^), ZIF-67 (78.31 mV dec^−1^), Co(OH)_2_ (78.73 mV dec^−1^), and commercial RuO_2_ (155.04 mV dec^−1^). Furthermore, the Tafel slope value of NiCo-LDH RDC@CNTs was within the scope of 60–120 mV dec^−1^, demonstrating that the chemical adsorption of OH^−^ step controls the OER kinetics (i.e., M + OH^−^ → M-OH + *e*^−^) [40,41].

This low overpotential and Tafel slope of NiCo-LDH RDC@CNTs were even comparable to commercial RuO_2_ and other recently reported works.

The extraordinary OER activity and satisfying OER kinetics together highlight the excellent OER performance of NiCo-LDH RDC@CNTs, which can mainly be ascribed to the following reasons from two aspects. First, the introduction of conductive CNTs can not only link the separate NiCo-LDH nanocages, but can also act as electron transfer highways, which can efficiently accelerate the electron conductivity of NiCo-LDH RDC@CNTs. As confirmed by the electrochemical impedance spectroscopy (EIS) Nyquist plots (Figure 5d), the NiCo-LDH RDC@CNTs displayed the smallest charge transfer resistance compared with other counterparts, indicating the fast charge-transfer capacity and promoted electron conductivity and OER kinetics. Furthermore, the specifically designed LDH nanosheet self-assembled nanocage architecture could supply sufficient and more accessible active sites for the OER. To estimate the reserve capacity of the active sites, the double-layer capacitance (*C*_dl_) tests were conducted by cyclic voltammetry (CV) in the non-faradaic region. As shown in Figure 5e and Appendix A, the NiCo-LDH RDC@CNTs exhibited a large *C*_dl_ value of 2.45 mF cm^−2^, which was much larger than that of ZIF-67@CNT (0.43 mF cm^−2^) and ZIF-67 (0.57 mF cm^−2^). Generally, the *C*_dl_ value of the electrocatalyst at the solid–liquid interphase is linearly dependent on the electrochemical surface area (ECSA). Thus, the higher *C*_dl_ value of NiCo-LDH RDC@CNTs implies the superior ECSA and more exposed catalytically active sites for the OER. Moreover, this special architecture can also provide robust structural and mechanical strength for the NiCo-LDH RDC@CNTs. To examine the long-term stability of the NiCo-LDH RDC@CNTs, continuous cyclic voltammetry (CV) scanning in the potential range of 1.3–1.5 V was performed. As shown in Figure 5f, after 2000 continuous cycles, the polarization curves of NiCo-LDH RDC@CNTs exhibited inappreciable attenuation in terms of overpotential and current density. To further probe the extraordinary OER stability of NiCo-LDH RDC@CNTs, the current–time (*i–t*) chronoamperometry test was conducted. As depicted in the inset of Figure 5f, after ultra-long continuous operation for over 27 h, negligible current retention could be observed, confirming the superior catalytic stability of NiCo-LDH RDC@CNTs. Moreover, the basic architecture units of NiCo-LDH RDC@CNTs were well maintained after thee stability test, while the Ni, Co, and O elements did not show significant erosion (Appendix A), verifying its robust structural and compositional stability. Considering the above-measured excellent OER activity, satisfactory OER kinetics, good electron transfer capacity, large exposed active surface area, it is reasonable to define NiCo-LDH RDC@CNTs as a promising substitute electrocatalyst for the OER.

Considering the extraordinary OER performance of NiCo-LDH RDC@CNTs, a rechargeable Zn–air battery was assembled to prove its practical application value. The basic configuration of the assembled battery is schematically illustrated in the inset of Figure 6a. The rechargeable Zn–air battery was assembled with NiCo-LDH RDC@CNTs and Pt/C (mass ratio of 1:1) as the air-cathode catalyst, Zn plate as the anode, and 0.2 M ZnCl_2_ + 6 M KOH mixed solution as the electrolyte. As shown in Figure 6a, the open-circuit voltage (OCV) of the NiCo-LDH RDC@CNTs + Pt/C based battery was measured to be about 1.45 V and could run stably for more than 1000 s. The charge/discharge polarization curves of the assembled battery showed a rapid current increasing trend as the cell voltage changed (Figure 6b). Moreover, according to the discharge curve, the NiCo-LDH RDC@CNTs + Pt/C based battery displayed a high power density of 189.0 mW cm^−2^ (Figure 6c). In addition, Figure 6d displays the galvanostatic charge–discharge curves of the as-assembled Zn–air battery at the current density of 5 mA cm^−2^. The charge and discharge voltage could be well maintained even after continuous operation for more than 200 cycles. These results further verify the application potential of NiCo-LDH RDC@CNTs in energy-related applications.

## 4. Conclusions

In summary, we developed a novel OER electrocatalyst composed of carbon nanotube interconnected NiCo layered double hydroxide rhombic dodecahedral nanocages with the assistance of a feasible zeolitic imidazolate framework (ZIF) self-sacrificing template strategy. Profited by the specifically designed LDH nanosheet self-assembled nanocage architecture and the CNTs’ interconnected features, the as-synthesized NiCo-LDH RDC@CNTs integrated a large exposed active surface area, multi-dimensional mass/electron diffusion channels, promoted electron transfer capacity, and robust structural stability, which collaboratively contributed to the excellent OER performance of NiCo-LDH RDC@CNTs. Electrochemical tests demonstrated that the designed NiCo-LDH RDC@CNTs displayed extraordinary OER activity with a low overpotential of 255 mV at 10 mA cm^−2^, satisfactory OER kinetics, and robust long-term stability (steady operation for over 27 h with negligible current fluctuation). Considering the above-measured excellent OER performance, we certainly believe that the as-prepared NiCo-LDH RDC@CNTs are a promising substitute electrocatalyst for the OER and sustainable energy-related systems.

## Figures and Tables

**Figure 1 nanomaterials-12-01015-f001:**
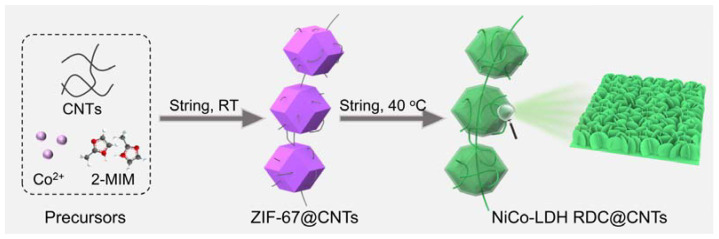
Schematic illustration for the synthesis of NiCo-LDH RDC@CNTs.

**Figure 2 nanomaterials-12-01015-f002:**
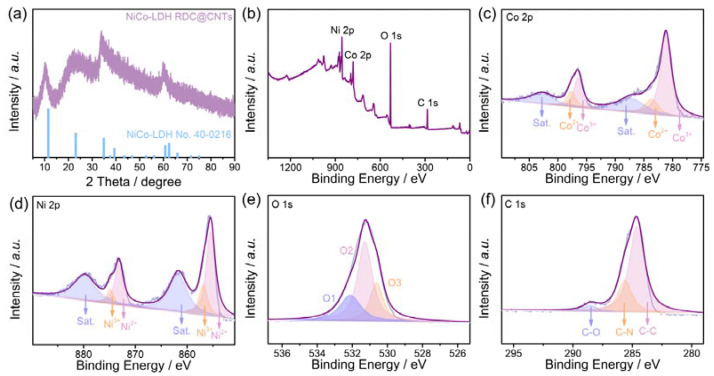
(**a**) XRD pattern; (**b**) survey scan XPS spectrum; high-resolution (**c**) Co 2*p*, (**d**) Ni 2*p*, (**e**) O 1*s*, and (**f**) C 1*s* XPS spectrums of NiCo-LDH RDC@CNTs.

**Figure 3 nanomaterials-12-01015-f003:**
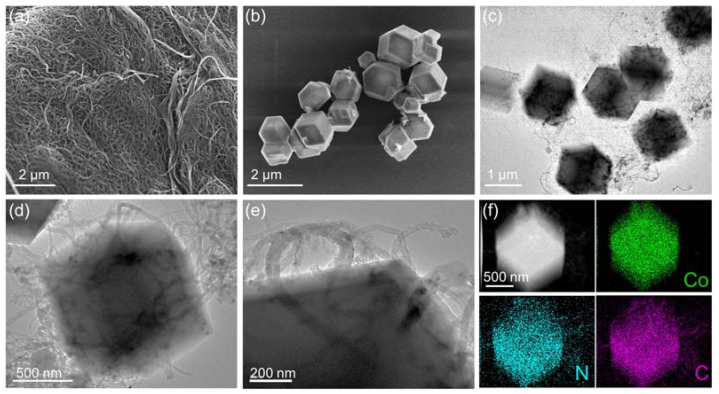
SEM images of (**a**) CNTs and (**b**) ZIF-67; (**c**–**e**) TEM images of ZIF-67@CNTs, (**f**) HADDF-STEM image, and corresponding EDS elements mapping profiles of ZIF-67@CNTs.

**Figure 4 nanomaterials-12-01015-f004:**
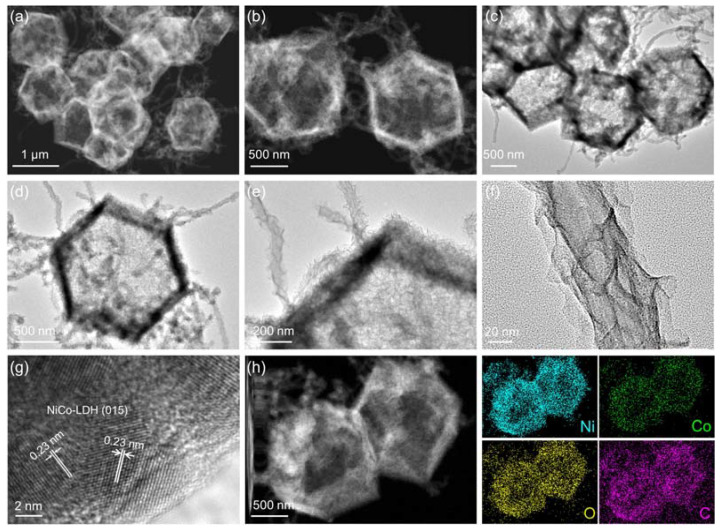
Morphology and structure characterization of NiCo-LDH RDC@CNTs: (**a**–**b**) HADDF-STEM images, (**c**–**f**) TEM images, (**g**) HRTEM image, and (**h**) HADDF-STEM image and corresponding EDS element mapping profiles.

**Figure 5 nanomaterials-12-01015-f005:**
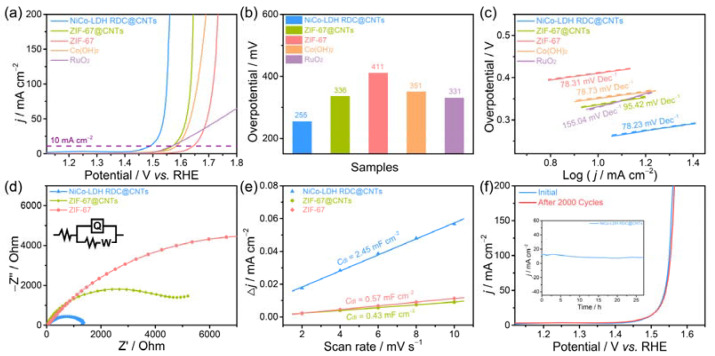
Electrocatalytic performance of different samples toward OER: (**a**) the OER polarization curves, (**b**) overpotentials at a current density of 10 mA cm^−2^, (**c**) Tafel plots, (**d**) fitted EIS Nyquist plots, and the inset shows the equivalent circuit diagram, (**e**) *C*_dl_ values of different samples, (**f**) the OER polarization curves of NiCo-LDH RDC@CNTs before and after 2000 cycles, inset shows the chronopotentiometry curve of NiCo-LDH RDC@CNTs.

**Figure 6 nanomaterials-12-01015-f006:**
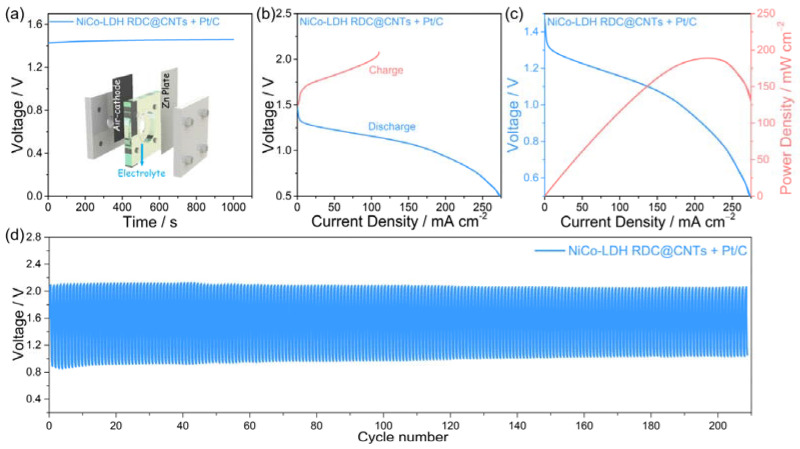
Zn–air battery performance of NiCo-LDH RDC@CNTs + Pt/C based battery: (**a**) OCV plots, and the inset shows the schematic illustration of Zn–air battery configuration. (**b**) The charge–discharge polarization curves; (**c**) discharge polarization curve and corresponding power density profile. (**d**) The galvanostatic charge-discharge curves of the Zn–air battery.

## Data Availability

The data presented in this study are available on request from the corresponding author.

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
