# Peer review of "Carbon Nanotubes Interconnected NiCo Layered Double Hydroxide Rhombic Dodecahedral Nanocages for Efficient Oxygen Evolution Reaction"

_nanomaterials, 2022, doi:10.3390/nano12061015_

Round 1

Reviewer 1 Report

This manuscript describes the preparation of a catalyst composed of a NiFe/carbon nanotube-LDH rhombic dodecahedral nanocages with a high OER activity. However, as cited in ref.26 in this manuscript, Dai et al. have already reported a high OER activity using a NiFe/carbon nanotube-LDH, and the OER activity of the present catalyst is comparable to that of Dai’s catalyst. Thus this reviewer does not consider that at the present form, the manuscript, the manuscript meets the criteria of this journal. Also, this reviewer suggests the authors to carry out the following studies and add the obtained results to the manuscript by comparing the data with those of previously reported papers.

  • ORR activity
  • Water oxidation test
  • Zn-air buttery preparation using the catalyst

Author Response

We would like to thank you review our manuscript. Based on your comments and suggestions, we have revised our manuscript carefully. All changes have been highlighted in the revised manuscript by formatting the text in YELLOW color. Detailed lists of revisions are listed in the attachment.

Reviewer 2 Report

The present manuscript on OER at NiCo layered double hydroxide rhombic dodecahedral nanocages is well written. However, research on this topic actually has done a lot. In addition, authors failed to present their data properly while a comparison with the resent finding is crucially needed. Some points should be addressed for considering further.

  1. As the best OER result among these catalysts has appeared upon addition of Ni precursor addition, it should be optimized by varying Ni precursor along with the reaction temperature.
  2. Although, it is clear in TEM that the sample has hollow structure, the BET surface area measurement is needed to verify it further.
  3. OER polarization curves should be compared with before and after stability test.
  4. The used catalyst in stability test should be checked by XPS and TEM to identify the role of N, Ni and/or Co as most active sites.
  5. The overpotential at 10 mA cm-2 and the Tafel slope should be compared with recent reports for better understanding of this catalyst’s extraordinary OER activity.
  6. Please give a picture of as assembled cell, if possible.

Author Response

(The authors gave the same response as above.)

Reviewer 3 Report

After examining the manuscript I am unable to conclude that the overall impacts are high enough to justify for publishing the manuscript in its present from at the journal “Nanomaterials”. But, after significant changes it may be considered for publication.

Specific comments on the manuscript:

  1. For the introduction part, a comprehensive review is needed. Authors have used the synthesized material for OER. Hence, the progress of OER with metal oxide, effect of nanostructure shape and size, porosity should be discussed by putting their numerical value.
  2. Please index XRD peaks in Figure . From XRD data it seems the sample is nanocrystalline which completely contradict with SEM images. Please recheck XRD data.
  3. Scale  in Figure 5d should be same in both axis. Please fit the EIS data and explain it.
  4. Authors should explain the results and origin of this interesting behavior of the samples. 

Author Response

(The authors gave the same response as above.)

Round 2

Reviewer 2 Report

I think, the manuscript is nicely corrected based on reviewer's comments. Therefore, this manuscript could be accepted in the current form.

Author Response

 Thank you very much again for your attention and kindness to improve our work.

Reviewer 3 Report

Authors have significantly improved the manuscript in the revised version. But I do not agree with the explanation of XRD result.

  1. Ni-Co LDHs may be crystalline and it confirms in SEM image. But XRD data shows only peak of CNT not of NI-Co-LDHs. Please check it.
  2. In EIS fitting, please provide the fitted data, equivalent circuit and outcome in the manuscript.
